# Physical Function and Association with Cognitive Function in Patients in a Post-COVID-19 Clinic—A Cross-Sectional Study

**DOI:** 10.3390/ijerph20105866

**Published:** 2023-05-18

**Authors:** Durita Viderø Gunnarsson, Kamilla Woznica Miskowiak, Johanna Kølle Pedersen, Henrik Hansen, Daria Podlekareva, Stine Johnsen, Christian Have Dall

**Affiliations:** 1Department of Physiotherapy and Occupational Therapy, Copenhagen University Hospital-Bispebjerg and Frederiksberg, 2400 Copenhagen, Denmark; christian.have.dall@regionh.dk; 2Neurocognition and Emotion in Affective Disorders (NEAD) Centre, Department of Psychology, University of Copenhagen-Mental Health Services, Capital Region of Denmark, 1172 Copenhagen, Denmark; kamilla.woznica.miskowiak@regionh.dk (K.W.M.); johanna.koelle.pedersen@regionh.dk (J.K.P.); 3Respiratory Research Unit and Department of Respiratory Medicine, Copenhagen University Hospital-Amager and Hvidovre, 2650 Hvidovre, Denmark; henrik.hansen.09@regionh.dk; 4Department of Rehabilitation Sciences and Physiotherapy, University of Antwerp, 2610 Antwerp, Belgium; 5Department of Respiratory Medicine and Infectious Diseases, Copenhagen University Hospital-Bispebjerg and Frederiksberg, 2400 Copenhagen, Denmark; daria.podlekareva.02@regionh.dk (D.P.); stine.johnsen.01@regionh.dk (S.J.); 6Department of Clinical Medicine, University of Copenhagen, 2200 Copenhagen, Denmark

**Keywords:** post-COVID-19 condition, long COVID, physical function, physical impairment, cognitive function, cognitive impairment

## Abstract

Patients with long-term health sequelae of COVID-19 (post-COVID-19 condition) experience both physical and cognitive manifestations. However, there is still uncertainty about the prevalence of physical impairment in these patients and whether there is a link between physical and cognitive function. The aim was to assess the prevalence of physical impairment and investigate the association with cognition in patients assessed in a post-COVID-19 clinic. In this cross-sectional study, patients referred to an outpatient clinic ≥ 3 months after acute infection underwent screening of their physical and cognitive function as part of a comprehensive multidisciplinary assessment. Physical function was assessed with the 6-Minute Walk Test, the 30 s Sit-to-Stand Test and by measuring handgrip strength. Cognitive function was assessed with the Screen for Cognitive Impairment in Psychiatry and the Trail Making Test-Part B. Physical impairment was tested by comparing the patients’ performance to normative and expected values. Association with cognition was investigated using correlation analyses and the possible explanatory variables regarding physical function were assessed using regression analyses. In total, we included 292 patients, the mean age was 52 (±15) years, 56% were women and 50% had been hospitalised during an acute COVID-19 infection. The prevalence of physical impairment ranged from 23% in functional exercise capacity to 59% in lower extremity muscle strength and function. There was no greater risk of physical impairment in previously hospitalised compared with the non-hospitalised patients. There was a weak to moderate association between physical and cognitive function. The cognitive test scores had statistically significant prediction value for all three outcomes of physical function. In conclusion, physical impairments were prevalent amongst patients assessed for post-COVID-19 condition regardless of their hospitalisation status and these were associated with more cognitive dysfunction.

## 1. Introduction

The coronavirus disease 2019 (COVID-19) has resulted in over 670 million cases and over 6.5 million deaths worldwide as of March 2023 [1]. With increasing numbers of patients who have survived the illness, it is necessary to attain a better understanding of the long-term health sequelae of COVID-19.

The World Health Organization (WHO) defines the long-term health complications as post-COVID-19 condition, which “…occurs in individuals with a history of probable or confirmed SARS-CoV-2 infection, usually 3 months from the onset of COVID-19 with symptoms and that last for at least 2 months and cannot be explained by an alternative diagnosis…” [2].

Reports of post-COVID-19 condition show that a substantial proportion (32–76%) of survivors of moderate to severe COVID-19 infection experience one or more symptoms in the long term (2–6 months) after diagnosis, where the most commonly reported symptoms include fatigue, general pain, malaise, muscle weakness, sleep disturbances, dyspnoea, chest pain, cognitive and mental health symptoms [3,4,5,6,7,8]. The evidence indicates that even in mild COVID-19 infection without the need for hospitalisation, individuals experience long-term manifestations from COVID-19 [8,9,10,11,12,13,14].

The prevalence of physical sequelae after COVID-19 infection ranges from 6.5–53.8% across studies [3,5,15,16,17] with most studies being based on follow-up after hospitalisation, with the follow-up time varying from 1 to 7.5 months from either symptom onset, diagnosis, hospital admission or discharge from hospital [16]. Studies show that patients experience reduced physical capacity, endurance and muscle strength and impaired mobility [5,9,15,16,18].

The International Classification of Functioning, Disability and Health (ICF) defines physical impairment as “…problems in body function or structure such as a significant deviation or loss” [19]. With this definition in mind, the underlying mechanisms and factors affecting patients’ physical function and the degree of impairment in relation to post- COVID-19 condition are complex, not comprehensibly understood and thus reflected in the current studies on the subject [3,16]. A systematic review points out that it can be difficult to untangle the sequelae caused directly by COVID-19 infection from those arising from related factors such as sequelae from hospitalisation due to severe illness [3]. Some studies suggest that COVID-19 infection can act as an immune trigger and the immune response may partly explain the persistent symptoms affecting the patients’ physical function [20,21]. Other studies emphasize the role of persistent dyspnoea [18], fatigue [22], post-exertional malaise [11,22], altered cardiorespiratory function [22], joint pain [4], muscle weakness [5,22] and deconditioning after bedrest [17] as contributing factors to functional impairment. This notion of the underlying mechanisms being multifactorial is also reflected in The National Institute for Health Care and Excellence (NICE) guidelines for managing the post-COVID-19 condition, which recommend that rehabilitation programs should be multidisciplinary and holistic [23].

Long-term persistent cognitive sequelae are prevalent in individuals after COVID-19 infection [7,11]. Some studies have found overlap and co-occurrence of symptoms of cognitive sequelae, breathlessness, psychological stress and reduced physical activity and function within the same population after COVID-19 infection [7,11,17]. It is suggested that cognition plays an integral role in most physical tasks and in physical function in general [24,25]. Although most of the studies on the subject include older populations (>65 years of age), these mechanisms can be thought to play a certain part regarding patients with post-COVID-19 condition. It can be hypothesised that both the cognitive and the physical system can be affected at once or that the COVID-19 infection affects one system the most, or maybe first, which then can impact the other system. This brings us to the question of a possible association between the physical and cognitive function in patients with post-COVID-19 condition, which is sparsely investigated in the literature.

Thus, the aim of this study was to assess the prevalence and pattern of physical impairments in patients referred to a post-COVID-19 outpatient clinic, and if the physical impairments differed between those previously hospitalised and non-hospitalised patients. Further, we aimed to assess if there was an association between physical and cognitive function in this large sample of patients. We hypothesised: (i) that patients would present with frequent physical impairments, (ii) that these impairments would be more frequent among previously hospitalised compared with non-hospitalised patients and (iii) that there would be a positive association between the patients’ physical and cognitive outcome scores.

## 2. Materials and Methods

### 2.1. Study Design and Participants

In this cross-sectional study, data were obtained from a post-COVID-19 outpatient clinic at Bispebjerg Hospital, in the Capital Region of Copenhagen, Denmark, from June 2020 to December 2021. Patients were consecutively enrolled and were eligible if they were ≥18 years of age and referred to the clinic either as part of a standard follow-up assessment after hospitalisation with COVID-19 or they were referred by their general practitioner due to unexpected, or complex and long-term symptoms (≥3 months) after COVID-19 infection.

This study was approved by the regional ethics committee (H-20035553), and all patients provided written and verbal informed consent.

### 2.2. Procedures

Patients attended the post-COVID-19 outpatient clinic where their physical and cognitive function was assessed within the same visit as part of a comprehensive clinical assessment, including examination by a medical doctor. Before the visit, the patients had completed a set of questionnaires by phone interview conducted by a nurse. Physical function was assessed by a physiotherapist and cognitive function was assessed by a neuropsychologist.

### 2.3. Assessment of Physical Function

To assess the patients’ functional exercise capacity, the 6-Minute Walk Test (6MWT) was performed according to a standardized protocol [26], where the patient was instructed to walk from one end to the other, turning around a cone placed at both ends, of a 20 m walkway at their own pace, while attempting to cover as much ground as possible in the allotted 6 min. Oxygen saturation, heart rate and dyspnoea, measured with the Borg dyspnoea scale [27], were assessed at rest immediately before and after the 6MWT.

Muscle strength and function in the lower extremities was assessed by patients performing the 30 s Sit-to-Stand Test where the patient was asked to rise to a full stand from a seated position, chair height 45 cm, as many times as possible in 30 s, with the arms across the chest [28].

As a surrogate measure for general peripheral muscle strength, the participants’ handgrip strength (HGS) was assessed with a handheld Saehan Hydraulic Hand Dynamometer following a standardized protocol [29]. In brief, the patient was instructed to sit comfortably in a standard chair with back support, and to rest their forearm of the dominant side on the armrest of the chair with their wrist just over the end of the armrest in a neutral position, thumb facing upwards. The patient was encouraged to squeeze the dynamometer for approximately 5 s as tightly as possible, while being encouraged by the physiotherapist. The highest grip strength measurement in kilograms, out of three attempts with the dominant hand, was read and recorded.

### 2.4. Assessments of Cognitive Function

Objective cognitive function was assessed with the brief (<20 min) cognition test battery Screen for Cognitive Impairment in Psychiatry—Danish Version (SCIP), version 3 [30], where a higher performance score indicated better cognitive performance. The test battery measured verbal learning and memory, working memory, verbal fluency, and processing speed.

Objective executive function was assessed with the Trail Making Test-Part B (TMT-B) [31,32], (score: number of seconds it took for the individual to complete the test) where a higher score indicated a poorer executive performance.

### 2.5. Patient-Reported Outcomes

Type and degree of respiratory symptoms were assessed using the COPD Assessment Test (CAT) [33] and Medical Research Council (MRC) Dyspnoea Score [34].

Health-related quality of life was assessed by obtaining the 5 Dimension 5 Level Quality of Life Questionnaire (EQ-5D-5L) [35].

The Post-COVID-19 Functional Status Scale (PCFS) [36] was completed to assess the patients’ subjective functional capacity.

Subjective cognitive functions were assessed with the Cognitive Failures Questionnaire (CFQ) [37].

Years of education were reported by the patients by asking them how many whole completed years of education they had, counting from primary school onward.

### 2.6. Other Outcomes

The patients’ comorbidity status was assessed using the Charlson Comorbidity Index (CCI) [38].

Time since COVID-19 infection was calculated from records of the date for positive SARS-CoV-2 PCR test, positive COVID-19 IgG titre or initiation of clinical symptoms of COVID-19 (for the patients that had not been tested), and the date of assessment in the outpatient clinic.

### 2.7. Statistical Analysis

Descriptive data were presented as means with standard deviation (SD) for continuous normally distributed data, as medians with interquartile range (IQR) for ordinal and not normally distributed data or numbers and percentages for categorical data.

The prevalence of physical impairment regarding submaximal functional capacity was tested by comparing the patient’s physical performance in the 6MWT to their expected performance calculated with regression-based formulas based on the patients age, sex, height and weight [39]. Cutoff value for physical impairment was set at <75% of expected performance.

The prevalence of physical impairment regarding muscle strength and function in the lower extremities and handgrip strength, respectively, was tested by comparing the patients’ physical performance in the 30 s Sit-to-Stand Test and HGS, respectively, to normative value intervals based on the patients’ age (divided into decades) and sex [40]. Cutoff values for physical impairment were set at > 1SD below expected performance for both tests [40].

Group comparisons of means of physical and cognitive function tests scores, respectively, comparing participants who had been hospitalised versus non-hospitalised, were conducted using independent t-tests.

Odds ratios (OR) for physical impairment, measured with the three physical function tests, in the hospitalised versus the non-hospitalised group were calculated using 2-by-2 tables. The chi-square test was applied to test for significance.

The association between physical and cognitive function was investigated with Pearson’s correlation analysis. Association was defined as very weak (r = ±0.19), weak (r = ±0.20 to ±0.39), moderate (r = ±0.40 to ±0.59), strong (r = ±0.60 to ±0.79) or very strong (r = ±0.80 to ±1) [41]. Linear regression with R-square measures were included to explain the amount of variation in the correlation.

To investigate the relationship between physical function outcomes and possible explanatory variables, multiple linear regression analyses were performed using physical function as the dependent variable and cognition, age, sex and education as independent variables.

All analyses were adjusted for age and sex with a significance level of 0.05 applied to all tests.

All statistical tests were conducted using STATA 17.0 (Texas, TX, USA).

## 3. Results

A total of 597 patients were assessed in the post-COVID-19 clinic in the inclusion period. Among these patients, 292 were included in the study as shown in the flowchart in Figure 1.

### 3.1. Participant Characteristics, Sociodemographic and Patient Reported Data

Table 1 presents the patients’ demographic, social and clinical characteristics stratified by their hospitalisation status in the acute phase of COVID-19 infection.

Fifty-six percent of the participants were female, and the mean age was 52 (±15) years. Age- and sex-adjusted analysis showed that compared to the non-hospitalised patients, the patients who had been hospitalised during their acute COVID-19 infection were older (*p* < 0.001), had a higher body mass index (BMI) (*p* < 0.001) and a higher degree of comorbidity (CCI; *p* < 0.001) (Table 1). Furthermore, the hospitalised group included more participants with asthma (*p* < 0.001) compared to the non-hospitalised group. Regarding working status, the hospitalised group had fewer individuals that were currently working and more who were retired (*p* < 0.022). In addition, there was a shorter time since acute COVID-19 infection in the hospitalised group (*p* = 0.001) and this group had a substantial overrepresentation of men compared to the non-hospitalised group (64% vs. 24%; *p* < 0.001) (Table 1).

All patient-reported outcome scores as well as years of education were similar across both groups. Regarding rehabilitation plans, a higher proportion of non-hospitalised patients had a rehabilitation plan made in the post-COVID-19 clinic (*p* = 0.001) and thus were referred to rehabilitation in a municipality setting (Table 1).

The missing values for the variables in Table 1 were distributed equally across the two groups.

### 3.2. Acute Severity for Hospitalised Patients

Assessment of the acute COVID-19 severity for the patients who had been hospitalised during their acute infection (*n* = 145) showed that the patients had been admitted for a median of 8 (IQR 5–14) days, 44.2% of the patients had received oxygen therapy, 58.2% had received high flow nasal cannula therapy and 11% had been intubated.

### 3.3. Physical and Cognitive Function

In total, 125 (43%) patients completed all five tests, i.e., all three physical tests and both cognitive tests. A total of 180 (62%) patients completed all three physical tests and 188 (64%) completed both cognitive tests, respectively. Table 2 shows the patients’ physical and cognitive function overall scores as well as the scores for the non-hospitalised and hospitalised group, respectively. The age- and sex-adjusted analysis showed that compared to the non-hospitalised group, the hospitalised group had statistically significantly lower muscle strength and function in the lower extremities (30 s Sit-to-Stand Test, *p* = 0.035) as well as lower functional exercise capacity (6MWT, *p* = 0.024). The mean handgrip strength was similar across the groups. Because of the notion that male sex has a significant positive impact on handgrip strength, the mean HGS score for males and females, respectively, are presented separately for this measure, solely for visual purposes. Regarding the patients’ cognitive function, the hospitalised group presented with a statistically significantly lower cognitive performance compared to the non-hospitalised group (SCIP-D total, *p* < 0.001), but the executive function performance was similar across the groups (TMT-B, *p* = 0.164).

### 3.4. Prevalence and Risk of Physical Impairment

The prevalence of physical impairment is shown in Table 3. The overall prevalence of physical impairment ranged from 23% in functional exercise capacity (6MWT) to 59% in lower extremity muscle strength and function (30 s Sit-to-Stand Test).

There were no statistically significant group differences regarding the risk of physical impairment, when having been exposed to hospitalisation, for any of the three physical function outcome measures (Table 3, OR (95%CI)). When exploring handgrip strength stratified by sex, there was still no statistically significant risk of impairment for either sex regarding having been exposed to hospitalisation (Table 3, OR (95%CI)).

The age- and sex-adjusted analysis showed that the patients in the non-hospitalised group classified in the normal area had statistically significantly higher performance scores in all three physical function outcome measures compared to those classified in the normal area in the hospitalised group. When handgrip strength was stratified by sex, the statistically significant higher performance scores only applied for females classified in the normal area (Table 3). The performance scores for the patients classified in the impaired area were similar across the non-hospitalised and hospitalised groups except for the handgrip strength measure, where the non-hospitalised group had a statistically higher mean score (Table 3).

### 3.5. Association between Physical and Cognitive Function

In Figure 2, the correlations between the patients’ physical and cognitive function are illustrated. All correlations were statistically significant. Functional exercise capacity (6MWT) correlated moderately with cognitive performance (SCIP) and executive function (TMT-B), respectively. Further, muscle strength and function in the lower extremities (30 s Sit-to-Stand Test) and handgrip strength (HGS), respectively, correlated weakly, but significantly, with both cognitive outcomes.

When stratifying by sex, the correlations remained similar for most outcomes. However, the correlations between handgrip strength (HGS) and both cognitive outcomes went from weak to moderate for men. This also applied for the men regarding the correlations between muscle strength and function in the lower extremities (30 s Sit-to-Stand Test) and cognitive performance (SCIP), and functional exercise capacity (6MWT) and executive function (TMT-B), respectively.

### 3.6. Physical Function and Possible Explanatory Variables

The multiple linear regression analyses presented in Table 4 show that the proportion of the variance for the three physical function tests, as dependent variables, explained by the deployed independent variables, ranged from 18% for muscle strength and function in the lower extremities to 56% for handgrip strength, respectively. SCIP and age had statistically significant explanatory prediction value for all physical function variables. For functional exercise capacity and for handgrip strength, respectively, sex was also a statistically significant explanatory variable (Table 4). When including TMT-B instead of SCIP in the regression analyses, the same pattern was seen, except that age lost its statistically significant predictor value for muscle strength and function in the lower extremities in this analysis.

## 4. Discussion

The main findings of this current study were that there was a high prevalence of physical impairment in this large sample of 292 patients assessed in a post-COVID-19 clinic >3 months after illness. Specifically, for functional exercise capacity, handgrip strength and muscle strength and function in the lower extremities, the prevalence was 23%, 36% and 59%, respectively. In contrast with our hypothesis, the risk of having physical impairments >3 months after an acute COVID-19 infection was similar regardless of whether the patients had been hospitalised. Moreover, we found weak to moderate significant associations between all investigated measures of physical and cognitive function in these patients. The multiple regression analyses showed that these associations were not confounded by age, sex, or years of education.

The high prevalence of physical impairment in our study is consistent with other studies of individuals with post-COVID-19 condition [3,15,42]. One large study with previously hospitalised patients found a prevalence of 20% for functional exercise capacity (6MWT), 22% for handgrip strength and 60% for isometric quadriceps strength at a 3-month follow-up after acute hospitalisation due to COVID-19 infection [42]. The prevalence was statistically significantly higher among patients with severe acute COVID-19 disease compared to moderate acute disease [42]. Regarding milder cases without the need for hospitalisation, there are substantially fewer studies on the subject and most of the existing studies present patient-reported symptoms [11,43,44,45]. One of these studies, with mostly non-hospitalised participants (86%), found a prevalence of 94% of patients reporting slight to moderate functional limitations (PCFS score of 2–3) more than 6 months after COVID-19 infection [45], whereas our study found a slightly lower prevalence of 73% reporting a score of more than 2 in the PCFS. Considering these results together in a bigger perspective, together with our findings of the prevalence of physical impairment being similar regardless of hospitalisation, it is apparent that the prevalence of both clinically assessed physical impairment and patient-reported functional limitation is substantial amongst patients in the long term after mild, moderate, and severe COVID-19 infection.

We opted for a cutoff of 75% of the expected performance in the 6MWT as a pragmatic approach to the fact that there was no consensus about what the precise cutoff should be [46]. For example, the beforementioned study [42] used a cutoff value of below 70% to determine physical impairment, where another study [47] proposed that values below 82% of the expected 6MWT performance could be considered abnormal. Therefore, the cutoff of 75% was as an attempt to balance the estimate of the prevalence of impairment in this outcome.

In terms of the cutoff values for determining the impairment of handgrip strength and muscle strength and function in the lower extremities, respectively, the set cutoff values for HGS and the 30 s Sit-to-Stand Test, respectively, were based on a new Danish normative reference material [40] derived from large (*n* = 1305–8342) Danish population studies including populations with the age span from 18 to >90 years [48,49,50,51]. This made them applicable to our sample containing a Danish population with a relatively broad age span. Noting that the 30 s Sit-to-Stand Test reference values, traditionally used for determining muscle strength impairment in the lower extremities, are based on an elderly American population (>60 years) [28], the new Danish material must be considered as a valuable upgrade in the process of valid classification of impairment in this area.

Other relatively new (2013–2016) international (British, Canadian, and German) studies have presented reference values for healthy, populations aged 6–90 years concerning handgrip strength [52,53,54]. Our reference material on handgrip strength [40] accounted for the patients’ sex and age, whereas the beforementioned studies also accounted for the patients’ height [52,54] and body weight [53], as these are factors known to affect handgrip strength. This could be considered a limitation in our study. On the other hand, our material closely matched the participants assessed in our study because the reference data had been developed in the country in which the participants lived.

When comparing the reference material for HGS used in our study to a large (*n* = 11.790) German population study [54], which found a critically weak grip at 1 SD below the age group specific means for males and females, our material had a slightly higher threshold for impairment in females (a difference of 0.4–0.8 kg at age 20–49 and 1.3–2.3 kg at age 50–90, respectively), young males (difference of 0.6 kg at age 20–29) and for the older males (difference of 1.3–2.2 kg at age 60–90). For males aged 30–39, the materials were similar and for the middle-aged males our material had a slightly lower threshold (difference of −0.4–−0.7 kg at age 40–59) for physical impairment regarding HGS. This could have led to under- or overestimation of the physical impairment for some of the age groups in our study. On the other hand, it could be argued that the differences were below what is considered to be the minimal detectible change score (5.0–6.5 kg) in HGS [55].

Interestingly, we found consistent correlations between all three physical function tests and the two cognition tests, the SCIP total score—a global cognitive performance measure spanning psychomotor speed, learning and memory and working memory—and the TMT-B score, a measure of executive function. These consistent associations between physical and cognitive sequelae of COVID-19 in our sample were remarkable. This might indicate common pathophysiological mechanisms underlying both symptom domains, perhaps through the common effects of mental fatigue and exhaustion. The 6MWT showed the most robust correlation with the cognitive tests, which is in line with what is known about factors reducing the distance walked in the test. In addition to female sex, shorter height and higher body weight, cognitive impairment is a contributing factor for reduced walking distance [56]. In this context, it is worth mentioning that the question of cognitive impairment is beyond the scope of this study. A comprehensive analysis of cognitive impairments has been published elsewhere [57]. The 6MWT evaluates the global and integrated responses of the systems involved in exercise [56], including elements of cardiopulmonary endurance, which generally is connected with neurovascular plasticity, neurogenesis and upregulation of neurotrophins that contribute to better brain health [58]. This connection between underlying physiological mechanisms may play a role in the robust correlation found between the patients’ functional exercise capacity and their cognitive function in our sample. The 6MWT can therefore arguably be considered a test of choice in the process of screening patients for post-COVID-19 condition, where patients experience physical and cognitive difficulties.

The clinical implications of this study touch upon the continued need for a multidisciplinary approach in the assessment and rehabilitation of patients with post-COVID-19 condition. After years of living with the pandemic and the changes it has brought to our society, most people now have had the infection and/or are vaccinated. Thus, COVID-19, including the post-COVID-19 condition, is, for the time being, an integrated part of our society, including the health care system. Based on this development, it is crucial for the health care system to ensure a reliable and sustainable basic assessment for patients experiencing long-term sequelae from COVID-19, for instance by including the 6MWT, which is easy, brief, and inexpensive to complete, in the initial screening process. This can facilitate the triage of patients to identify those with more advanced assessment and rehabilitation needs requiring an often more costly, multidisciplinary, and holistic approach in relevant sectors and settings, depending on disease severity and comprehensiveness of the needed assessment and treatment. These perspectives are supported by the findings of a rapid systematic review [59], where international care models for people with post-COVID-19 sequelae include a coordination unit and primary care pathways as well as access to multidisciplinary rehabilitation and specialised medical services.

### 4.1. Strengths

A strength in our study was the large sample including both previously hospitalised and non-hospitalised patients.

In the classification of physical impairment, we used normative reference material applicable to Danish adults of a broad age span.

The outcome measures included for physical function (6MWT, 30 s Sit-to-Stand Test and HGS) are well validated and because of their generic properties they are highly suitable for this heterogeneous population. These three outcome measures are also commonly used in other studies assessing physical function in patients with post-COVID-19 condition [60,61,62,63]. Both the 6MWT and 30 s Sit-to-Stand Test are recommended in a recent multidisciplinary guideline for the management of patients with post-COVID-19 condition [64], which increases the generalisability of this study. Regarding the cognitive outcome measures, the SCIP test is highly sensitive to cognitive deficits in general and specifically sensitive to the cognitive sequelae of COVID-19 [65].

### 4.2. Limitations

This was an observational study, which entailed that causality could be drawn from the results.

The patients included in the study were in the process of assessment due to unexpected, or complex and long-term symptoms (≥3 months) after a COVID-19 infection, and some of the previously hospitalised patients were assessed based on a routine referral after hospitalisation, without necessarily having long COVID symptoms. This fairly inconsistent inclusion method might potentially influence the generalisability of the study.

In the 6MWT and the 30 s Sit-to-Stand Test, respectively, patients were only given one trial, which is a deviation from the protocol since there is a known learning effect in these two tests. This might have resulted in patients not being able to perform to their maximal value with the consequence of a possible overestimation of the prevalence of physical impairment in this sample.

Although we included a large sample, the lack of data completeness was a limitation. The PCFS, 6MWT and 30 s Sit-to-Stand Test were added to the test battery later in the process after the post-COVID-19 clinic was initiated, which explained the missing data in these outcomes compared to, e.g., HGS that was a part of the test battery from the beginning.

## 5. Conclusions

Our study showed that the prevalence of physical impairment in this large sample of patients assessed in a post-COVID-19 outpatient clinic, regarding functional exercise capacity, handgrip strength and muscle strength and function in the lower extremities, was substantial. The patients that were hospitalised during their acute COVID-19 infection did not have increased risk of having physical impairments >3 months after their acute infection compared to patients without the need for hospitalisation. Moreover, we found significant associations between physical and cognitive function in these patients.

These findings highlight the fact that patients referred to post-COVID-19 clinics are challenged in multiple domains simultaneously and that the acute severity of the COVID-19 infection does not necessarily dictate the severity or prevalence of physical impairment. However, further research on the causalities of physical impairment in this patient population is warranted.

## Figures and Tables

**Figure 1 ijerph-20-05866-f001:**
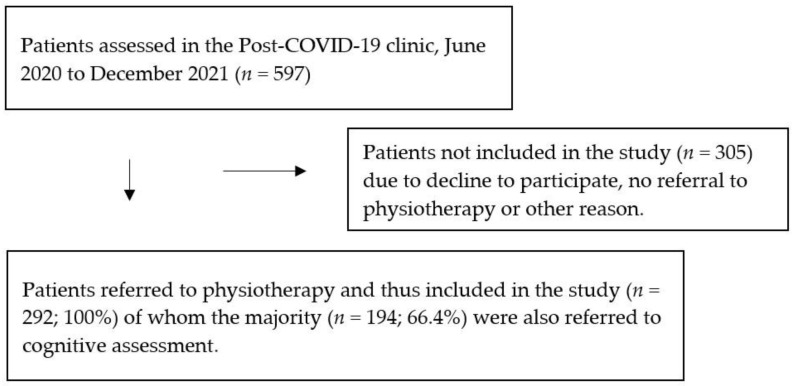
Flowchart over inclusion of patients.

**Figure 2 ijerph-20-05866-f002:**
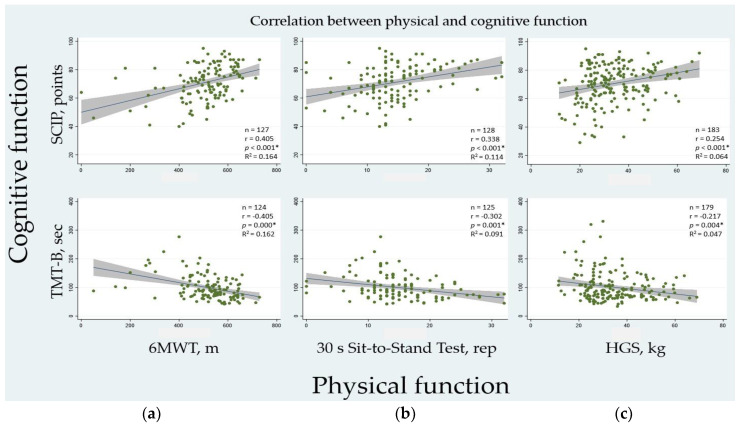
Scatterplots of correlation between physical and cognitive test scores. Green dots are individual cognitive and physical test scores for the respective measures. The blue lines are fitted lines. The gray areas represent the 95% CI. Statistically significant correlations are denoted with a *. (**a**) 6MWT values on both upper and lower *x*-axis, SCIP values on the upper *y*-axis and TMT-B values on the lower *y*-axis; (**b**) 30 s Sit-to-Stand Test values on both upper and lower *x*-axis, SCIP values on the upper *y*-axis and TMT-B values on the lower *y*-axis; (**c**) HGS values on both upper and lower *x*-axis, SCIP values on the upper *y*-axis and TMT-B values on the lower *y*-axis. For all three physical function tests as well as for the SCIP, a higher score equals a better performance. For TMT-B, a lower score equals a better performance. Abbreviations: rep, repetitions; 6MWT, the 6-Minute Walk Test; m, metres; HGS, handgrip strength; kg, kilograms; SCIP-D, Screen for Cognitive Impairment in Psychiatry—Danish Version 3; TMT-B, the Trail Making Test-Part B; sec, seconds.

**Table 1 ijerph-20-05866-t001:** Participant characteristics, sociodemographic and patient reported data.

Sociodemographic Data	Overall *n* = 292	Non-Hospitalised *n* = 147	Hospitalised *n* = 145	*p*-Value
Age, years, mean (SD)	51.9 (15.2)	45.7 (14.3)	58.2 (13.3)	<0.001 *
Sex				
Women, *n* (%)	164 (56.2)	112 (76.2)	52 (35.9)	<0.001 *
Men, *n* (%)	128 (43.8)	35 (23.8)	93 (64.1)	
Race or ethnic group (missing, *n* = 6)				
Caucasian, *n* (%)	212 (74.1)	120 (81.6)	92 (63.5)	<0.001 *
Other, *n* (%)	74 (25.9)	22 (15.0)	52 (35.9)	
BMI (missing, *n* = 3)				
kg ^x^ m^−2^, mean (SD)	27.3 (12.1)	25.2 (3.9)	29.4 (5.9)	<0.001 *
> 30 kg ^x^ m^−2^, *n* (%)	69 (23.9)	14 (9.5)	55 (37.9)	0.181
Smoking (missing, *n* = 41)				
Never smoker, *n* (%)	143 (57.1)	76 (51.7)	67 (46.2)	0.659
Current smoker, *n* (%)	16 (6.4)	12 (8.2)	4 (2.8)	
Previous smoker, *n* (%)	92 (36.7)	39 (26.5)	53 (36.6)	
Time since cessation, median years (IQR)	15.0 (5.1–30.0)	10.0 (4.0–25.0)	21.5 (11.0–40.0)	0.002 *
Work status (missing, *n* = 53)				
Currently working, *n* (%)	162 (68.6)	101 (68.7)	61 (42.1)	<0.022 *
Out of work, *n* (%)	34 (14.2)	15 (10.2)	19 (13.1)	
Retired, *n* (%)	43 (18.0)	6 (4.1)	37 (25.5)	
Comorbidities				
CCI, median (IQR) (missing, *n* = 61)	2.0 (1.0–3.0)	1.0 (0.0–2.0)	3.0 (1.0–4.0)	<0.001 *
Asthma, *n* (%) (missing, *n* = 40)	46 (18.3)	15 (10.2)	31 (21.4)	<0.001 *
COPD, *n* (%) (missing, *n* = 39)	8 (3.2)	3 (2.0)	5 (3.5)	0.532
Time since COVID-19 infection, days, mean (SD) (missing, *n* = 102)	217.2 (111.5)	261.4 (115.6)	166.1 (148.9)	<0.001 *
Patient reported measures				
Education, years, mean (SD) (missing, *n* = 95)	15.3 (4.0)	15.6 (3.9)	14.9 (4.1)	0.343
MRC score, mean (SD) (missing, *n* = 59)	2.2 (0.8)	2.1 (0.7)	2.2 (1.0)	0.290
CAT score, mean (SD) (missing, *n* = 55)	13.9 (7.3)	14.7 (7.1)	13.0 (7.4)	0.678
PCFS score pre-COVID-19, median (IQR) (missing, *n* = 121)	0.0 (0.0–0.0)	0.0 (0.0–0.0)	0.0 (0.0–0.5)	0.656
PCFS score post-COVID-19, median (IQR) (missing, *n* = 117)	2.0 (1.0–3.0)	2.0 (2.0–3.0)	2.0 (1.0–3.0)	0.914
EQ-5D-5L index, median (IQR) (missing, *n* = 87)	0.84 (0.71–0.93)	0.86 (0.65–0.95)	0.82 (0.73–0.91)	0.580
EQ-5D-5L VAS, median (IQR) (missing, *n* = 87)	70.0 (50.0–70.0)	66.0 (50.0–80.0)	70.0 (50.0–75.0)	0.561
CFQ total, mean (SD) (missing, *n* = 94)	38.1 (17.5)	40.6 (17.4)	35.3 (17.4)	0.959
Rehabilitation plan (yes), *n* (%)	69 (23.6)	47 (32.0)	22 (15.2)	0.001 *

Any statistically significant difference between the non-hospitalised and the hospitalised group is denoted with a *, significance level: *p* < 0.05. Analyses are age- and sex-adjusted. Abbreviations: SD, standard deviation; BMI, body mass index; IQR, interquartile range; CCI, Charlson Comorbidity Index; COPD, chronic obstructive pulmonary disease; MRC, Medical Research Council Dyspnoea Score; CAT, COPD Assessment Test; PCFS, Post-COVID-19 Functional Status Scale; EQ-5D-5L, the 5 Dimension 5 Level Quality of Life Questionnaire; VAS, Visual Analog Scale; CFQ, the Cognitive Failures Questionnaire.

**Table 2 ijerph-20-05866-t002:** Physical and cognitive function scores.

Physical Function Tests	*n*	Overall	*n*	Non-Hospitalised	*n*	Hospitalised	*p*-Value
30 s Sit-to-Stand Test, rep, mean (SD)	181	14.3 (6.0)	100	15.1 (6.5)	81	13.4 (5.3)	0.035 *
6MWT, m, mean (SD)	180	489.5 (138.7)	99	507.9 (121.5)	81	467.0 (155.0)	0.024 *
HGS, kg, mean (SD)	237	33.0 (12.0)	124	31.2 (10.2)	113	34.9 (13.7)	0.904
HGS, females, kg, mean (SD)	134	26.0 (7.9)	94	27.2 (6.3)	40	25.3 (10.8)	0.835
HGS, males, kg, mean (SD)	103	41.3 (11.6)	30	40.2 (12.2)	73	44.0 (9.5)	0.992
**Cognitive tests**							
SCIP-D total, points, mean (SD)	196	70.4 (14.1)	105	75.2 (10.8)	91	64.8 (15.4)	<0.001 *
TMT-B, seconds, mean (SD)	189	100.3 (50.2)	103	88.8 (44.5)	86	113.9 (53.3)	0.164

Any statistically significant difference between the non-hospitalised and the hospitalised group is denoted with a *, significance level: *p* < 0.05. Analyses are age- and sex-adjusted. Abbreviations: SD, standard deviation; rep, repetitions; 6MWT, the 6-Minute Walk Test; m, metres; HGS, handgrip strength; kg, kilograms; SCIP-D, Screen for Cognitive Impairment in Psychiatry—Danish Version 3; TMT-B, the Trail Making Test-Part B.

**Table 3 ijerph-20-05866-t003:** Prevalence of physical impairment and mean scores in the overall, non-hospitalised and hospitalised group, respectively. *p*-values apply for the comparison of means in the non-hospitalised vs. hospitalised group. Odds ratios (OR) apply for physical impairment in the non-hospitalised vs. the hospitalised group.

	Overall	Non-Hospitalised	Hospitalised	*p*-Value	OR (95%CI)
Functional exercise capacity (6MWT), *n* = 180					
Normal, *n* (%) m, mean (95%CI)	139 (77) 540.2 (526.7–553.7)	79 (80) 546.7 (529.8–563.7)	60 (74) 531.6 (509.2–554.1)	0.031 *	1.38 (0.65–2.95)
Impaired, *n* (%) m, mean (95%CI)	41 (23) 317.6 (268.4–366.9)	20 (20) 354.5 (285.6–423.3)	21 (26) 282.6 (209.8–355.4)	0.632	
Muscle strength and function in the lower extremities (30 s Sit-to-Stand Test), *n* = 181					
Normal, *n* (%) rep, mean (95%CI)	75 (41) 18.8 (17.6–20.0)	41 (41) 20.2 (18.5–22.0)	34 (42) 17.1 (15.5–18.6)	0.049 *	0.95 (0.51–1.82)
Impaired, *n* (%) rep, mean (95%CI)	106 (59) 11.2 (10.4–12.1)	59 (59) 11.5 (10.4–12.6)	47 (58) 11.0 (9.7–12.3)	0.766	
Handgrip strength (HGS), *n* = 237					
Normal, *n* (%) kg, mean (95%CI)	152 (64) 37.7 (35.8–39.5)	83 (67) 34.2 (32.0–36.4)	69 (61) 41.8 (39.1–44.5)	<0.003 *	1.29 (0.73–2.28)
Impaired, *n* (%) kg, mean (95%CI)	85 (36) 24.7 (22.9–26.5)	41 (33) 25.3 (23.0–27.7)	44 (39) 24.1 (21.3–26.9)	<0.011 *	
Handgrip strength (HGS)—Females, *n* = 134					
Normal, *n* (%) kg, mean (95%CI)	86 (64) 30.2 (45.3–49.5)	63 (47) 29.7 (28.3–31.0)	23 (17) 31.6 (28.0–35.1)	<0.001 *	1.50 (0.65–3.43)
Impaired, *n* (%) kg, mean (95%CI)	48 (36) 20.3 (18.5—22.1)	31 (23) 22.2 (20.4–24.0)	17 (13) 16.8 (13.4–20.2)	0.063	
Handgrip strength (HGS)—Males, *n* = 103					
Normal, *n* (%) kg, mean (95%CI)	66 (64) 47.4 (45.3–49.5)	20 (67) 48.5 (45.0–52.0)	46 (63) 46.9 (44.3–49.6)	0.268	1.17 (0.44–3.24)
Impaired, *n* (%) kg, mean (95%CI)	37 (36) 30.4 (28.0–32.9)	10 (33) 35.0 (31.7–38.3)	27 (37) 28.7 (25.9–31.6)	0.070	

Any statistically significant difference between the non-hospitalised and the hospitalised group is denoted with a *, significance level: *p* < 0.05. Analyses are age- and sex-adjusted. Abbreviations: OR, odds ratio; 6MWT, the 6-Minute Walk Test; m, meters; 95%CI, 95% confidence interval; rep, repetitions; HGS, handgrip strength; kg, kilograms.

**Table 4 ijerph-20-05866-t004:** Multiple linear regression with the three physical function tests as dependent outcomes and SCIP, age, sex, and education years as independent variables, respectively.

	Coefficient	95%CI	*p*-Value
Functional Exercise Capacity (6MWT)			
R^2^ = 0.27			
SCIP	3.08	1.38–4.78	<0.001 *
Age	−2.15	−3.39–−0.92	0.001 *
Sex (female)	−46.92	−84.67–−9.15	0.015 *
Education years	−1.95	−6.95–3.05	0.442
Muscle strength and function in the lower extremities (30 s Sit-to-Stand Test)			
R^2^ = 0.18			
SCIP	0.12	0.04–0.21	0.006 *
Age	−0.09	−0.16–−0.03	0.006 *
Sex (female)	−1.69	−3.64–0.27	0.090
Education years	−0.07	−0.33–0.19	0.573
Handgrip strength (HGS)			
R^2^ = 0.56			
SCIP	0.17	0.07–0.27	0.001 *
Age	−0.21	−0.29–−0.13	0.000 *
Sex (female)	−16.48	−18.85–−14.11	0.000 *
Education years	−0.02	−0.35–0.30	0.881

Any statistically significant explanatory variable is denoted with a *, significance level: *p* < 0.05. Abbreviations: 95%CI, 95% confidence interval; R^2^, R-square; 6MWT, the 6-Minute Walk Test; HGS, handgrip strength; SCIP-D, Screen for Cognitive Impairment in Psychiatry—Danish Version 3; TMT-B, the Trail Making Test-Part B.

## Data Availability

The raw data are not available to the public according to rules of the Danish Data Protection Agency.

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
