# Peer review of "Physical Function and Association with Cognitive Function in Patients in a Post-COVID-19 Clinic—A Cross-Sectional Study"

_ijerph, 2023, doi:10.3390/ijerph20105866_

Round 1
Reviewer 1 Report
Review of: Physical function and association with cognitive function in patients in a post covid-19 clinic – a cross-sectional study.
Manuscript ID ijerph-2315513
In the introduction the authors state clearly the aims and hypotheses of their study. In my opinion their study is adequately designed to address the aims.
The following are my comments (major and minor comments combined):
Abstract: The abstract is clear and includes all the relevant information.
Introduction: The introduction is clear and generally reads well. However, please note, in line 52 and also in line 78, when discussing cognitive impairment that frequently, in all sorts of papers, that the term “cognitive impairment” is used wrongly. It is frequently used when there is only a statistically significant difference between two groups although both fall within the cognitively normal range. This is not appropriate. Please look at this in the paper you reference. Otherwise the text in misleading, unclear or possibly wrong.
Chapter 2.4., Assessment of cognitive function: There are 3 parallel versions of the Danish version of the SCIP I believe. Which version was used here? Are there Danish norms? And where those norms used in this study?
Chapter 2.6., Statistical Analysis: It is stated that you control for age and sex. What about education? Controlling for education is vital when comparing groups on cognitive tests. Also, I believe education has been shown to be associated with Covid-19 outcome in other aspects, i.e. the likelihood of long Covid.
Results: Figure 1: It is always helpful to include % to help the reader. Thus one quickly sees the proportion that participates at each stage of the study.
Chapter 3.3., Physical and cognitive function: the group that underwent cognitive and physical testing ranges from being 43% to 63% of the original sample. It is a clear disadvantage not knowing the demographics on those groups. Are the groups different from the larger sample shown in Table1, possibly indicating a selection bias? And as the authors do not control for education I have a hard time making sense of the cognitive results presented in Table 2. This is a clear limitation of the study, making it really hard to interpret all cognitive data, including those presented in Figure 2.
Discussion/Conclusion:
The fact that there is no discussion of education level in a study on cognition post-Covid-19 is a major limitation making all evaluation of the manuscript very difficult.
English language and proofreading: generally good, however, one more reading of the manuscript is advised and some errors were spotted, e.g., in line 201 a hyphen is missing (patient-reported), in lines 194 and 200 there is an extra period in parentheses (table 1.). I will not proofread fully nor indicated other errors in the manuscript.
Reviewer 2 Report
First of all, I would like to thank the authors and the editor for the possibility of reviewing such an interesting study that delves into a subject that still needs to be studied further, such as the effects of COVID-19 at different levels of the population, especially patients suffering from long COVID or permanent COVID.
Firstly, and in reference to the formal elements of the article, it presents a correct structure and complies with all the elements that guarantee its quality. It is worth highlighting the topicality of all the references used in the preparation of this scientific article, which is reflected both in the introduction and in the discussion, fundamentally.
The introduction is pertinent, coherent with the development of the article and, as I have already mentioned, with the use of current references that help to introduce the reader to the subject. It is a correct introduction and conforms to the journal's quality standards.
In the section on materials and methods, the authors correctly explain the characteristics of the participating sample as well as the processes of evaluation of cognitive function and physical or motor aspects. Likewise, the tests presented by the authors are pertinent and are scientifically valid. Similarly, the statistics used by the researchers are in line with the objectives pursued in this research.
The results section is presented in a clear and orderly manner, and the figures and tables presented by the authors facilitate the understanding of the results obtained. Perhaps even information is presented that does not contribute value to the research, such as some of the information presented in Table 1: Characteristics of the participants Characteristics of the participants, socio-demographic data and data reported by the patients. Nevertheless, this table provides extra information that better situates the reader.
Likewise, the authors relate physical and cognitive function very well, and also present very interesting material that helps to see and interpret the results, such as figure 2.
Finally, as I have already mentioned, the discussion and conclusion are pertinent. The discussion is based on interesting and novel research, supporting the authors' work.
For all these reasons, the quality of the work presented is high, both in terms of the research itself and the way it has been presented in the scientific article.
Reviewer 3 Report
Thanks for the opportunity to read this interesting manuscript.
Brief summary
The aim of the study was to assess the prevalence of physical impairment, regarding functional exercise capacity, handgrip strength and muscle strength and function in the lower extremities, of patients with post-COVID. Additionally, the study investigated association of physical impairment with cognition. The results show that the prevalence of physical impairment ranged from 21% in functional exercise capacity to 59% in lower extremity muscle strength and function. Furthermore, there was no higher risk of physical impairment in hospitalized patients compared with the non-hospitalized patients during acute SARS-CoV-2 infection. The results detected weak to moderate association between physical and cognitive impairments. The large sample size and the use of objective, valid measurements are strengths of the study. The authors reported study strengths and limitations.
General and specific comments
The manuscript is relevant for the field of research in post-COVID, long-term impairments, and aftercare. It is also represented in a well-structured manner. The manuscript is well written, and the methods are adequate. The relevant literature is cited.
However, some changes might further improve the manuscript.
In the article the term long-COVID (e.g., line 20) and post-COVID (e.g., line 33) were used. As patients referred to an outpatient clinic ≥ 3 months after acute infection, the term post-COVID in comparison to the NICE guideline could be preferred throughout the text.
Abstract
Next to the correlation analyses the regression analyses could be also mentioned in the abstract.
Introduction
The introduction is well written. Nevertheless, it should be explained why the question of a possible association between physical and cognitive function in patients with post COVID-19 should be investigated (line 81). What are the underlying hypotheses/ mechanism that are suspected?
Please use uniform tenses (e.g., line 48 and 79).
Materials and Methods
As shown in figure1 a large sample size of 292 post-COVID patients were recruited in the current study. But yet, the missing values of each measurement are very differently (e.g., MRC: missing: n=59, PCFS: missing n=117, 6MWT: missing n=112). Can you give more information about these circumstances (missing, drop-outs) please? Are the missing values distributed in the same way between both groups (hospitalized/non-hospitalized patients)? Maybe, the sample size of the reported data could be harmonized with the missing values (suggestion: only data of patients with completed datasets in physical parameters could be reported).
The CCI is mentioned in table 1 but not described in the section „Methods“.
As part of the statistical analysis independent t-test were used for group comparisons (hospitalized versus non-hospitalized patients, table 1). As some of your data are ordinal or not normally distributed, non-parametric tests should be preferred.
Can you explain please, why the cut off value for physical impairment compared to the 6MWT score was set at <75 % of expected performance and not >1 SD or more?
Results
It may be interesting if there are differences in physical and cognition function test scores between hospitalized patients with and without ICU. In line with the literature, it could be expected that are differences in patients with ICU and non-hospitalized patients.
In tables 1-3 the values of the test statistics (T or F-value) and effect sizes should be also reported. Please check the given sample size in table 3 in each line carefully (e.g.: 6-MGT Normal: overall n=142, non-hospitalised n=79, hospitalised n=60 à 79+60=139). In figure 2 the number of the examined subsamples and the units of each measurement should be added.
Discussion
The discussion is well written. The relationship of physical and cognitive functions in post-COVID could be discussed taking into consideration further theories and current study results. What are possible reasons for the different correlation values (6MWT vs. HGS)?
References
Reference-style should be checked (e.g., reference 50: capital letters). DOI citations should be provided for all references.
Thanks again for the opportunity to read this interesting manuscript.
Reviewer 4 Report
The paper addresses some relevant issues in evaluating and managing the so-called Post-COVID condition (prevalence of impairments and correlation between impairments...). The methodology is clearly presented, as well as the results. The initial severity in hospitalized patients was graded as O2 therapy, high-flow cannula, or intubation (lines 214-215). A more detailed description of severity (e.g. the four levels proposed by Wu et al. Clin. Infect Dis 2020, 71) would be helpful, as well as an analysis of the possible correlations between the initial severity level and the long-term impairments. Sex differences in the correlations found between physical and cognitive function (line 262-268) could also be investigated.
In the discussion, (line 377-379) the findings of a recent review ( Dugas M, Stefan T, Langlois L, Skidmore B, Bhéreur A, and LeBlanc A. (2021). Care Models for Long COVID – A Rapid Systematic Review. SPOR Evidence Alliance, COVID-END Network) could be mentioned
Round 2
Reviewer 1 Report
I have no comments at this stage
Reviewer 3 Report
Thanks for the opportunity to read the revised manuscript.
All my comments have been noted and responded by the authors. The quality of the revised manuscript has been improved.